# Preparation of Ligand Brush Nanocapsules for Robust Self-Controlled Antimicrobial Activity with Low Cytotoxicity at Target pH and Humidity

**DOI:** 10.3390/pharmaceutics14020280

**Published:** 2022-01-25

**Authors:** Eunjoo Koh, Yong Taek Lee

**Affiliations:** Department of Chemical Engineering, Faculty of Chemical Engineering, Kyung Hee University, Deogyeong-daero, Giheung-gu, Yongin-si 17104, Korea; ejkoh@khu.ac.kr

**Keywords:** controlled drug release, self-antibacterial activity, ligand brush nanocapsule, pH and humidity response, cytotoxicity

## Abstract

This study prepared nanocapsules (NCs) with excellent self-controlled antimicrobial activity at pH 6–7 and humidity 45–100%, conditions in which most bacterial and fungal strains thrive. The nanocapsule substrate (NC@SiO_2_) was 676 nm in diameter, and the ligand-grafted capsule (NC@SiO_2_-g-MAA) was 888 nm. The large surface area and outer ligand brush of the NCs induced a rapid, self-controlled antibacterial response in the pH and humidity conditions needed for industrial and medical applications. Ligand-brush NCs containing an anionic antimicrobial drug had a rapid release effect because of the repellent electrostatic force and swelling properties of the ligand brushes. Controlled release of the drug was achieved at pH 6 and humidity of 45% and 100%. As many carboxylic acid groups are deprotonated into carboxylic acids at pH 5, the NC@SiO_2_-g-MAA had a high negative charge density. Carboxylic acid groups are anionized (–COO^−^) at pH 6 and above and push each other out of the capsule, expanding the outer shell as in a polymer brush to create the release behavior. The surface potential of the NC intermediate (NC@SiO_2_-MPS) was −23.45 [mV], and the potential of the capsule surface decreased to −36.4 [mV] when the MAA ligand brushes were grafted onto the surface of the capsule intermediate. In an antimicrobial experiment using *Escherichia coli*, a clear zone of 13–20 mm formed at pH 6, and the *E. coli* was eradicated completely at pH 6 and pH 7 when the humidity was 100%.

## 1. Introduction

Polymeric, pH-responsive nanocapsules (NCs) [1,2,3,4,5] have been applied in a variety of biotech and environmental processes, such as pharmaceuticals [6,7,8,9,10], antibacterial packing materials, enzymes [11] and cosmetics. Temperature-sensitive polymers with a low critical solution temperature undergo phase separation, changing from hydrophilic to hydrophobic at certain temperatures. For example, hydrophilicity is retained at the normal temperature of the human body, 37 °C, but it changes to hydrophobicity at higher temperatures (about 42 °C), causing the nanoparticles to break through interactions among them and leading to instantaneous drug release. That method cannot be used with non-biodegradable material that does not decompose in the body. On the other hand, pH-sensitive polymers can release drug agents in response to pH changes without external help [12,13]. Change in the pH of blood and cells does occur, but this alteration from 7.4 is small and does not exceed ±0.04. Cells use endocytosis to absorb certain particles and form endosomes whose pH (6.0) is lower than that of the cytosol (7.15). In addition, the pH around cancer tumors is lower than that of the surrounding tissue because of the organic acid produced by cancer cell metabolism. In polymer nanoparticles, pH differences can cause phase shifts that prompt aggregation or expansion of the nanoparticle surface to induce drug release [14,15,16,17,18]. Therefore, effective self-antibacterial activity in the presence of rapid pH changes can be obtained by using ligand- and pH-sensitive functional groups on NCs [12,19,20]. Triclosan is a typical disinfectant material commonly used as a germicide/microbicide/fungicide. As drug carrier systems have developed to reduce the side effects in non-target sites through use of ligands, functional groups, and pH signals, triclosan has gained therapeutic efficiency. As a typical pH-responsive material, methalic acid (MAA) is widely used as a drug carrier that can be swollen/collapsed at particular pH values via ionization of its –COOH groups [21,22,23,24,25,26]. A smart pH-responsive polymer can be used to fabricate a multifunctional nanocomposite that acts as a drug container and carrier in vivo and in vitro [27,28].

The objectives of this study are fast and immediate antibacterial effect and excellent biocompatibility. We have sought out antibacterial activity and biocompatibility on the NCs. *E. coli* was picked on the antibacterial effect, and also biocompatibility was verified by the cytotoxicity test of the epithelial cells. Furthermore, we prepared pH-sensitive NCs via Pickering encapsulation technology [29,30,31,32]. These NCs release their cargo above pH 5 when the humidity is high, conditions in which most bacteria, including *Escherichia coli* and fungi, are active. In other words, we designed humidity- and pH-sensitive ligand brush–grafted [33,34,35,36,37] NCs [5,38,39,40] for use as a drug carrier. The grafted ligands serve as an outer shell to carry and control the release of an antibacterial drug (triclosan). The NC substrate [41,42,43,44] was modified with humidity- and pH-sensitive ligand brushes by applying 3-(trimethoxysilyl)propyl methacrylate (MPS) [45,46] to produce NC@SiO_2_-MPS intermediates that were further modified into ligand-brush NC nanospheres by grafting on the carboxylic group of MAA [47,48]. Anionic triclosan was used as a model drug, and controlled release was achieved at pH 4, pH 6 and pH 7. Moreover, we defined the mechanism for the controlled release of a negatively charged drug from the anionic outer-shell NCs [49]. The size of the NCs was controlled to 1.0 μm to ensure the proper surface area. Further antibacterial testing was performed to investigate the pH sensitivity of the anionic drug delivery system and determine its self-antifungal and -antimicrobial activities and its cytotoxicity toward *E. coli* at different pH and humidity levels. The developed self-healing antibacterial NCs can be used in various fields such as medical disposal antibacterial patches, infectious disease protective clothing, hemostatic patches, water filters, export packaging materials (high humidity areas), food and industrial packaging materials.

## 2. Experimental

### 2.1. Materials

Tetraethyl orthosilicate (TEOS), ammonium hydroxide (28~30%), polyvinylpyrrolidone (PVP, MW = 40,000), 3-(trimethoxysilyl)propylmethacrylate (MPS), sodium dodecylbenzenseulfonate, poly(methalic acid, sodium salt) solution (MAA), *N*,*N*′-methylenebisacrylamide (BIS), pentanol, sodium citrate, ammonium persulfate (APS), ammonium hydroxide, absolute ethanol, triclosane (CAS No. 638-67-5), and buffer solutions (pH 4, 6) were purchased from Sigma-Aldrich (Yongin-si, Korea). The cell resource is human lung epithelial cells (A549 cells) that were cultured in the Specially Materials Lab. of Kyung Hee University.

### 2.2. Preparation of Self-Healing Antibacterial NCs

#### 2.2.1. Synthesis of Self-Assembly NCs

PVP (2.5 *g*) was dispersed in 25 mL of pentanol and stirred at 1 g force for 30 min at 60 °C. Immediately after stirring, the sample was transferred to a prepared bath (60 °C, 1 L) and sonicated for 1 h. After sonication, the PVP-pentanol mixture was placed in a centrifuge with water (2.5 g), sodium citrate solution (0.016 g of sodium citrate in 243.16 μL of water), ethanol (1.8936 g), and ammonium hydroxide (250 μL) and vortex-stirred for 2 min at 408× *g* force. Next, 1 mL of TEOS was added and continuously vortexed for 2 min at 408× *g* force and then overnight [23]. The supernatant was collected, rinsed with ethanol, centrifuged in the same manner, and dried overnight. The dried NCs were then ground to powder, as shown in Figure 1a [50,51,52,53,54,55].

#### 2.2.2. Surface Modification

Ethanol (100 mL) and MPS (0.5 mL) were added and stirred at 6× g force for 24 h at 25 °C to coat the NC surfaces (100 mg). After centrifugation at 6523× *g* force for 10 min, the sample was washed with 20 wt.% ethanol. The NCs were dispersed in solution (50 mL of water, 5 mg of sodium dodecyl benzene sulfonate) and stirred for 1 h at 6 *g* force, after which MAA solution (200 mg) and BIS (15 mg) were added. Then, the mixture was bubbled with N_2_ gas for 30 min, 10 mg of APS was quickly added [47,56], and the liquid was stirred at 6× *g* force and 75 °C for 4 h. After centrifugation at 6523× *g* force for 10 min, the sample was washed with distilled water and dried at 40 °C for 24 h, as shown in Figure 1b,c.

#### 2.2.3. Modification of pH-Sensitive Ligand-Brush NCs

As described in Figure 1, two 50 mg samples of nanospheres were dispersed in 10 mL of pH 4 and 6 buffer, for a total of four samples. To those samples, 10 mg of triclosan was added and stirred at 1 g force for 24 h at 25 °C in a water bath. After centrifugation at 408× *g* force for 10 min and drying at 40 °C for 24 h, the prepared NCs were dispersed in buffer (pH 4 and pH 6) at 50 mg/10 mL. The mixture was stirred continuously at 1× *g* force and 37 °C for 24 h. After centrifugation for 10 min at 408× *g* force, the supernatant was collected, and the pellet was dried at 40 °C for 24 h and stored (Figure 1c,d). The humidity- and pH-sensitivity of the self-antimicrobial NCs occurred through the following steps. The MPS modification induced a negative charge on the NC surface due to the equilibrium of the Si–OH group. The ionized silanol group (Si–OH) was crosslinked by the oxygen bridge of the MAA, which induced agglomeration on the grafted ligand brushes on the NC surface. The ligand-brush NCs had a suitable surface area and a release mechanism easily controlled by the swelling of the ligand brushes. All the grafted, hybrid, self-assembled NCs exhibited an excellent surface area that enhanced the loading capacity for the drug and affected the NCs’ antimicrobial activity. The NC@SiO_2_ was modified with humidity- and pH-sensitive brushes by applying MPS to prepare NC intermediates (NC@SiO_2_-MPS) that were modified into ligand-brush NC@SiO_2_-g-MAA nanospheres by grafting them with the carboxylic group of MAA, as shown in Figure 1c,d. The C=O bonds and Si–O–Si bonds of the polymer formed rapidly at high temperature and dissolved in the solvent before the polymer solidified, as shown in Figure 1d. However, if the crosslinked polymer is slowly added dropwise at a mild temperature (50 °C), capsules of uniform size with a smooth surface can be obtained, and agglomeration of the polymer and excessive tangling of the Si–O molecular chains can be prevented.

### 2.3. Antimicrobial Evaluation with E. coli

The antimicrobial activity of the NCs was determined by the disk diffusion test with *E. coli* [57,58,59,60]. The antimicrobial activity values were measured at pH 4 and pH 6 on a filter paper disk (control sample (C.S.)), NCs with no drug, and NCs with triclosan. All of the NC plates and the C.S. had a diameter of 10 mm. The bacterial strains were cultivated at 37 °C and 45% humidity for 12 h. After incubation, the bacterial solution was transferred evenly across the surface of an agar plate with a diameter of 90 mm in a sterile environment. For the 45% humidity condition, the concentration of growth medium (LB) in the agar plate was 0.08 g/L. The prepared 10-mm plates containing 55 mg of NCs were placed on top of the agar plates, and the C.S. and NC plates were placed in an incubator at 37 °C for 24 h. Then, the clear zone around each NC plate was measured to determine the diameter of inhibition. For the 100% humidity condition, the optical density of the cell suspension was determined by using a Synergy HTX multi-mode reader (BioTek, Winooski, VT, USA) to measure the concentration of *E. coli*. This test was repeated three times.

### 2.4. Cytotoxicity Test with Lung Epithelial Cells

The sample was diluted, and A549 human lung epithelial cells were dispensed (3.12 × 10^5^ cells in 10 mL of medium). The dispensed cells were cultured for 24 h, and then the sample was exposed to the test material/solution for 5 min under sonication. The samples were cultured for an additional 24 h, and then cell proliferation was observed with an optical microscope. The three test liquids were buffer solution (pH 6) for the control sample, and pristine NCs and drug-loaded NCs in buffer solution (pH 6). For the cytotoxicity test, when the number of surviving cells survives more than 85% of the initial number of cells, it can be analyzed that there is no cytotoxicity and good biocompatibility. The concentration of drug extracted from the capsule was 100 μg/mL, and cell adhesion activity was compared with the control sample.

### 2.5. Characterization

The morphological structures of the prepared nanofiber membranes were observed with a field emission scanning electron microscope (SEM, LEO SUPRA 55, Carl Zeiss, Oberkochen, Germany) and transmission electron microscopy (TEM, JEM-2100F (Tokyo, Japan, JEOL)), and the surface elemental compositions of the NCs were analyzed by X-ray photoelectron spectroscopy (XPS, K-Alpha (Thermo Electron, Billerica, MA, USA)). The particle size was measured using a particle size analyzer (Otsuka (ELSZ-2000ZS), Chiyoda, Japan). Particle sizes and zeta potentials were measured on the NC surfaces at 25 °C in deionized water. We used the conversion equation for zeta potential (mV) proposed by Smoluchowski and calculated it using Contin to indicate the surface electrons on the nanoparticles. The surface area was calculated using the Brunauer, Emmett, and Teller (BET, BELSORP-Max, BEL, Osaka, Japan) method. Surface chemical components were characterized using a Fourier transform infrared spectrometer (FT-IR; Spectrum One System, PerkinElmer, Waltham, MA, USA) over a wavelength range of 500–4000 cm^−1^, according to the shift in chemical absorption peaks. The controlled release of the drug agent was confirmed using UV-Vis spectrometry (CARY 300 Bio, Santa Clara, CA, USA). The A549 cells were tested using a cell viability assay kit (EZ-Cytox, Dogen, Seoul, Korea), and live cells were captured using an optical microscope (Olympus, CKX41, Hamburg, Germany).

## 3. Results and Discussion

### 3.1. Surface Morphology

Ligand-brush NCs have a suitable surface area and a release mechanism easily controlled by the swelling of the ligand brushes. All our grafted, hybrid, self-assembled NCs exhibited an excellent surface area that maximized the loading capacity of the drug and affected their antimicrobial activity. Figure 2a shows the surface formation and surface composition at a reaction temperature of 40 °C, which induced mild chemical reactions between the Si–O–Si and C=O and produced round, uniformly sized NCs. If the crosslinked polymer is slowly added dropwise at a mild temperature (40–50 °C), capsules of uniform size with a smooth surface can be obtained (Figure 2a,b). At 65 °C and 80 °C, NC encapsulation was prevented by agglomeration of the polymer and excessive tangling of the Si–O molecular chains (Figure 2c,d). The pH-sensitive NCs capable of storing and transporting drugs were prepared at different temperatures to confirm their uniformity, size, and surface shape by SEM (Figure 2). NC formation was performed between 40 °C and 80 °C to confirm the shapes at different temperatures. Capsules prepared at 50 °C showed a smooth surface and a relatively uniform size (Figure 2a,b) because the C=O and Si–O–Si bonds of the polymer form rapidly at 40–50 °C and are dissolved in the solvent before the polymer solidifies.

The surface morphology, particle size, and surface potential of the NCs were evaluated at each modification step and differed with the formation conditions. Figure 2a shows entangled ligand brushes caused by inconsistent target humidity and pH conditions. Figure 2(a1) shows the surface morphology and shape of the NCs used as supports. The NC@SiO_2_ formed by the Pickering method produced a weak bumpy pattern on the surface due to the adsorption and reaction of the polymer molecules. The ligand brushes were grafted onto the NC@SiO_2_-g-MPS by reacting them with the carboxylic group of MAA. Therefore, the NC@SiO_2_-MPS shows a raspberry-like surface because the hydrogen bonds have the silanol group (Si–OH) and (C=O)–OH structure (Figure 2(a2)). This NC@SiO_2_-MPS intermediate sphere can be combined with MAA. At that step, the (C=O)–OH bond of the NC@SiO_2_-MPS reacts with the carboxyl group (–COOH) of the MAA, leading to self-assembly of the NC@SiO_2_-g-MAA. The raspberry shape was also formed by the ligand-brush grafting and resulted in the –COO^−^ groups of MAA shown in Figure 2(a3). The preparation yield of the nanocapsules was 45%, as calculated by Equation (1) [61]:(1)$$\mathrm{Preparation}\;\mathrm{yield}\;\left(\%\right)=\frac{\mathrm{Nanocapsule}\;\mathrm{weight}}{\mathrm{Theoretical}\;\mathrm{weight}\;\mathrm{of}\;\mathrm{all}\;\mathrm{components}}\times 100$$

### 3.2. Analysis of Surface Components and Surface Area

As the capsules prepared under each condition used silica and polymer, spherical capsules were clearly distinguished in the TEM analysis. As shown in Figure 3, the surface composition, shape and size, and area of the NCs prepared at 40 °C were confirmed by SEM, EDS, and BET measurements, respectively, revealing well-formed NCs. A relatively uniform spherical capsule was observed by TEM: Figure 3(a1) shows the smooth shape of the surface of the NC@SiO_2_ substrate via Si–O–Si and C=O bonding. The EDS microanalysis was performed to provide qualitative determinations of the elemental composition. Figure 3(a4) shows the SEM image of the shape and size of the capsule surface, and the EDS mapping analysis of NC@SiO_2_ confirmed the presence of Si and O, as shown in Figure 3(a2,a3). The Si–O–Si bonds were the major constituents of the capsule surface. The shell elements of the NCs contained Si, C, and O, as shown in Figure 3(b2–b4). The surface area of the NC@SiO_2_ was 347.03 cm^2^/g, an increase of 3470 times compared with the surface area of the 1.0 g spherical capsule (Figure 3(a5)).

Figure 4b show the XPS and XRD patterns of the NCs and nanocomposites. All the intense peaks correspond with the surface elemental composition. The constituents of the NC surface were confirmed by the XPS analysis, as were the chemical bonds of the Si group and the specific polymer peaks. As the XPS analysis results show, the atomic percentages (At. %) of Si, N, O and C were 21.25, 4.07, 42.27 and 31.71, respectively, as shown in Figure 4a. The X-ray powder diffraction pattern of the prepared NCs shows an amorphous peak at 2θ = 21.8° caused by the equivalent Bragg angle marks. Martinez et al. (2006) prepared amorphous SiO_2_ by the sol-gel procedure. In their work, a Si–O–Si peak was centered at 2θ = 23°. The peak can be shifted to lower 2θ values depending on the molar ratio (R) of water to APS (R = 5 or 11.66). The XRD analysis indicated a crystal structure, which Figure 4b shows as one broadened XRD peak for amorphous silica centered at a 2θ value close to our measurement. The pure Si peak appearing at 28.5° is caused by chemical interactions between the silanol group (Si–O) and the carboxyl group (–COOH), and the peak of the Si was shifted by polymerization according to the oxygen (O) element. The Si–O–Si bond was found at 23°.

### 3.3. Analysis of Zeta Potentials and NC Sizes

Controlled drug release can be initiated when the ligand brushes on the NC surface produce a repellent electrostatic force. We investigated the electric potential of NCs at each modification step and observed the zeta potential and mobility, as shown in Table 1. The anionic zeta potential value of the NCs was further increased by the carboxyl groups of the MAA polymer. The zeta potential was negative, indicating an increase in negative charges on the particle surface due to equilibrium of the zeta potential of Si–O–Si, C=O, and –COO^−^ on the NC surfaces. The surface electron value of the NC@SiO_2_ was −37.43 because of the Si-O groups in the TEOS, and the zeta value increased after MPS modification (−23.45 mV). The MAA surface modification decreased the surface zeta potential to −36.41 as a result of the –COO^−^ groups of MAA [62,63]. That negative charge produced higher surface energy. The ionized silanol groups (Si–O) could be crosslinked by the oxygen bridge of the MAA and induce agglomeration of the ligand-brush chemical structure on the NC surfaces. As seen in Table 1, the mean particle diameter of the NC series was calculated with the standard deviation (S.D.) as 676.3 (±119.6) nm for NC@SiO_2_ and 761.2 (±100.1) nm for the NC@SiO_2_-MPS modified NCs, and the mean diameter of the surface-grafted NCs was 887.8 (±141.2) nm. The mean diameter of the NC increased under the influence of hydrogen bonding and brushes grafted to the NC surface. The uniformity of the NC diameter was excellent, producing a uniform thickness when used as a coating or a spray, and the amount of drug loading was relatively constant, allowing consistent antibacterial and bactericidal activity at any area of the sample.

### 3.4. Analysis of Drug Loading and Controlled Release of NCs

The FT-IR spectra in Figure 5 show surface modification, drug loading, and drug-controlled release at different pH conditions (pH 4 and 6). In this study, loading tests of the anionic triclosan were carried out at pH 6. The peaks corresponding to triclosan at pH 4 indicate that triclosan was not released in that condition, whereas the triclosan-loaded NCs demonstrated controlled released in pH 6 buffer solution due to the repellent electrostatic force shown in Figure 5a. In the triclosan spectra, the hydroxyl group (–OH) peak occurred at 3312 cm^−1^, C–Cl absorption bands were observed in a frequency range of 722–570 cm^−1^, and two strong CH wagging bands were observed for the CH_2_Cl group in the 1300–1150 cm^−1^ region. The spectra for the pristine NCs and ligand-brush NCs without triclosan loading are shown in Figure 5b. The –COO^−^ component of the MAA was observed on the ligand brushes grafted onto the NCs, and a peak corresponding to Si–O–Si was not imaged after the grafting modification. On the contrary, the drug-loaded NCs had absorption bands at 3450 cm^−1^ (OH) and 1630 cm^−1^ (H_2_O). On the grafted NC surfaces, SiO_2_ was present at 1080 cm^−1^ (Si–O–Si), 800 cm^−1^ (Si–O–Si) and 965 cm^−1^ (Si–OH), as shown in Figure 5a. The intensity of the Si–O–Si peak decrease is due to the ligand brushes on the surface of the NC@SiO_2_-MPS. The MAA polymer might induce a negatively charged electron density on the capsule surface. After modification with MPS, a new absorption band at 1705 cm^−1^ was observed and assigned to the vibration of C=O from MPS. Thus, the NC@SiO_2_ was successfully functionalized for drug loading. When the anionic antibacterial drug was loaded onto the NCs at pH 6, specific peaks were identified for each modification, allowing us to observe all the changing components.

### 3.5. Dynamic Swelling Behavior According to pH and Humidity

#### 3.5.1. Mechanism of Controlled Release

Among the polymers whose ionic properties change with pH, MAA, which we used as the ligand-brush shell, can be charged with anions at pH 5 and above. We analyzed the controlled release performance of drug-loaded NCs at pH 4, pH 6 and pH 7, and found that controlled release of the antibacterial drug occurred at pH above 5. Therefore, we designed a humidity- and pH-sensitive self-antimicrobial release system that is very sensitive under acidic conditions. In the release experiment for that agent, the capsules loaded under pH 6 were subjected to release testing at pH 4 and pH 6. As a result, we confirmed that the drug agent was released only above pH 6, as shown in Figure 6, because the anionic MAA ligand brush was pulled out of the anionic drug molecules by electro-repulsive forces as the entangled grafting polymer chains opened above pH 6. The NCs were influenced in different pH ranges due to deprotonation of the carboxylic acid groups, which gave rise to electro-repulsive forces between the carboxylate anions and the anions of the agent. In contrast, the entangled ligand-brush polymer chains did not expand and retained the entangled polymer chains, so the antibacterial drug loaded into the ligand brushes was not released at pH 4.

This controlled drug release can be explained by MAA in different pH conditions. pK_a_ indicates the dissociation of an acid, whereas pH indicates the acidity or alkalinity of a system, with a smaller pK_a_ value indicating a stronger acid [21,22,23,24,64]. The pK_a_ value of the pristine MAA polymer was ~4.8 at neutral pH, which means that it almost entirely deprotonated, making it an anionic polymer. The carboxyl groups are ionized to be electrically negative. A large number of carboxylic acid groups in MAA were deprotonated as carboxylic acid at a pH above 5, resulting in a high negative-charge density in the NCP@SiO_2_-g-MAA, which can cause swelling. At pH 6, most of the carboxylic acid groups of MAA were ionized as –COO^−^ anions and repelled each other, causing the polymer ligand-brush outer shell to swell and the network to open for controlled drug release. By contrast, at pH 4, the outer shell of the NCs did not swell due to the formation of hydrogen bonds among the carboxylic groups of the MAA. Second, the intermolecular interactions between the Si–O–Si groups from the MAA produced an electrostatic attraction or intermolecular hydrogen-bonding interaction among them at pH 4, hindering drug diffusion (Figure 6b). On the contrary, electrostatic repulsion occurred at pH 6 and accelerated the controlled release and diffusion of the agent from the NCs. The controlled release mechanism of the drug is depicted in Figure 6c. As ionization increases, the swelling behavior is governed by the relaxation mechanism of the polymer chain, which explains why the MAA ligand brushes swell above pH 6.0. When the pH is 6.0, q has a value close to 2, indicating that the solvent penetration mechanism is relaxation-controlled transport. On the contrary, when the pK_a_ value is acidic (pH 4.0 in this experiment), there is no repulsive force between the ligand brushes because ionization of the MAA ionization group does not occur, so the mechanism of controlled release is not affected by Fickian release.

#### 3.5.2. Dynamic Swelling Behavior According to pH and Humidity Response

To examine the dynamic swelling behavior of the hydrogel according to the pH, 500 mg of dry NCs were immersed in pH 4 and pH 6 buffer solutions for 8 h. The swollen NCs were then taken out of the buffer solutions. We measured their mass and calculated the mass swelling ratio, q, using Equation (2).
(2)$$q=\frac{W_{s}}{W_{d}}$$
where q is the weight swelling ratio, $W_{s}$ is the mass of the NCs after wet swelling, and $W_{d}$ is the dry mass of the NCs before swelling. Table 2 shows the change in the mass swelling ratio (q) of NCs soaked in buffer solutions at pH 4.0 and pH 6.0 for 8 h. The amount of water absorbed by the polymer network was greater when the pH was 6.0 than when the pH was 4.0. After 1 h in pH 4 buffer solution, the q value was 1.000, but in the pH 6 buffer solution, it was 2.011. This result can be explained by the fact that the hydrophilic network is increased by ionization of the carboxylic acid groups in the MAA repeat unit in the hydrogel at a pH higher than pK_a_ 5, which allows it to absorb a relatively large amount of moisture. Nevertheless, swelling of the weak MAA ligand brushes occurred even at pH 4.

At pH 5 or higher, the electrostatic repulsion between negatively charged groups in the NCs caused the ligand brush portion of the surface to expand rapidly and release the drug agent. The ionization of the carboxyl groups present in the MAA changed with the surrounding pH environment, and so selective drug agent release occurred as the pH changed, as shown in Figure 7. NCs capable of selective drug agent release were prepared using pH-switched polymers by ionizing the carboxyl groups present in MAA according to the surrounding humidity. The pH-sensitive NCs showed hydrophobicity at pH 4; at pH higher than that, the carboxyl groups of the MAA ionized and became electrically negative (pH 5 corresponds to the pK_a_ of MAA). At pH 6.0, the electrostatic repulsion between the negatively charged groups caused the ligand-brush portion of the MAA to expand rapidly, releasing the loaded antibacterial drug. As a result, the triclosan attached to the ligand was released, and triclosan was identified after its controlled released in both the pH 6 and pH 7 buffer solutions, confirming that controlled release occurred through the influence of pH, as shown in Figure 7a.

We evaluated the controlled drug release using drug-loaded NCs in pH 4 and pH 6 solutions, as shown in Figure 7b. For the controlled-release NCs, the adsorption of pH 6 DML buffer solution components at 3400 cm^−1^ and 3350 cm^−1^ corresponded to the –OH and –NH groups, respectively. However, the controlled-release NCs in pH 4 buffer preserved the specific peaks of triclosan (Figure 7b), indicating that controlled release did not occur at pH 4. As shown in Figure 7 for the NCs at pH 6, the MAA C=O absorption band was observed at 1709 cm^−1^, and the bands at 1500 cm^−1^ could be assigned to the stretching vibration absorption of C=O–C and the asymmetric stretching vibration absorption of –COO^−^ anion groups caused by grafting MAA onto the surface of the NC intermediate (NC@SiO_2_-MPS). These results demonstrate the chemical reactions between the (C=O)–OH of MPS and the carboxyl group of MAA that were generated when grafting the ligand brushes. In the controlled-release test at pH 6, the triclosan attached to the ligand was released, and no more drug material was identified, confirming that controlled release occurred through the influence of pH, as show in Figure 7. The drug release curve shows whether the drug was completely released at pH 6 and pH 7.

### 3.6. Self-Antimicrobial Activity

As described in Figure 8, the entangled MAA brushes grafted onto the hybrid NCs swelled only in certain humidity and pH environments, in which the increase of energy in the hydrogenated capsular surface produced electrostatic repulsion with the anionic drug. The self-antimicrobial activity of the drug-loaded NCs was tested at pH 4, pH 6 and pH 7. At pH 4, no clear zone of *E. coli* formed, as shown in Figure 8a. However, the pH 4–loaded NCs produced a 13-mm clear zone, and the pH 6-loaded NCs produced a 15–19 mm clear zone at 45% humidity, as shown in Figure 8b. In addition, the density change in *E. coli* was evaluated at pH 4, pH 6 and pH 7 and 100% humidity for 24 h. As shown in Figure 8c, the initial *E. coli* density with pristine NCs (without drug loading) was 0.117 (OD), which increased to 23.2 (OD) after incubation for 24 h at pH 6, indicating a greater than 198-fold growth of *E. coli*. The *E. coli* density with the drug-loaded NCs increased from 0.117 to 0.7 (OD) at pH 4, decreased about 3-fold at PH 6, and decreased from 0.105 to 0.1 (OD) at pH 7. Meanwhile, the *E. coli* density with the drug-loaded NCs decreased from 0.117 to 0.08 (OD) at pH 6 and increased slightly, from 0.105 to 0.3 (OD), at pH 7, an approximate 2.9-fold increase. Finally, all *E. coli* bacteria were killed after 24 h of exposure to the drug-loaded NCs in pH 6 and pH 7 conditions with 100% humidity, illustrating controlled release of the anionic antibacterial drug at the target pH. The weak swelling behavior of the MAA ligand brushes caused by moisture resulted in the release of the drug, which inhibited the growth of *E. coli*, a result similar to the swelling behavior seen in the NCs in the pH 4 buffer solution.

### 3.7. Cytotoxicity Testing

We next examined the numbers of live cells after cultivating A549 cells in various samples, as shown in Table 3 and Figure 9. The number of live cells that adhered to the flask before exposure and the number of live cells after cultivation for 24 h were quantified; cytotoxic tests were performed by determining the number of adherent cells and examining images of live cells. In Table 3, the cell adhesion activity was calculated using Equation (3).
(3)$$\mathrm{Cell}\;\mathrm{adhesion}\;\mathrm{activity}\;\left(\%\right)=\left(\frac{\mathrm{Number}\;\mathrm{of}\;\mathrm{cell}\;\mathrm{adhesion}\;\mathrm{after}\;\mathrm{exposing}\;\left(24\;h\right)}{\mathrm{Number}\;\mathrm{of}\;\mathrm{cell}\;\mathrm{adhesion}\;\mathrm{before}\;\mathrm{exposing}}\right)\times 100$$

The number of live cells in the control sample was 79, which increased to 188 after 24 h. In experiments using pristine NCs, the number of living cells increased from 70 to 98. With drug-loaded NCs, the number of living cells increased from 115 to 162. With the pristine, non-drug-loaded NCs, the cell proliferation rate was 237%, whereas the cell proliferation rate of the drug-loaded NCs was approximately 140%. Thus, the cell proliferation rate of the drug-loaded NCs was less than 97% of that in the control sample, but it was still 140%, indicating no cytotoxicity. Figure 9 shows images of dead and living cells after exposure to the indicated samples.

## 4. Conclusions

This work has demonstrated humidity- and pH-responsive NCs prepared with ion changeable polymer brush. The anion-modified NCs were formed as a container and loaded with an anionic antibacterial drug by the anionization of the grafted ligand brushes. The surface area (347.03 cm^2^/g) of our NCs was increased using a nanoscale capsule and copolymer ligand brushes. That large surface area contained a high density of the antibacterial drug, and drug release was easily controlled by the swelling of the ligand brushes. The grafted, anionic ligand-brush copolymer has an obvious positive influence on the negatively charged antibacterial drug. The diameter of the substrate NC@SiO_2_ was 676.3 (±119.6) nm, which increased to 887.8 (±141.2) nm upon grafting of the ligand brushes. We also defined the mechanism of the controlled drug release. The self-antimicrobial NCs showed a controlled release in target pH and humidity conditions. In the antimicrobial experiment using *E. coli*, a clear zone of 13–20 mm formed at pH 6 and 45% humidity, and all the *E. coli* was killed at pH 6 and pH 7 at 100% humidity. The density of *E. coli* decreased from 0.117 to 0 (OD) at pH 6 with 45% humidity and from 0.105 to 0 (OD) at pH 6 with 100% humidity, demonstrating self-antimicrobial activity at target humidity and pH conditions.

## Figures and Tables

**Figure 1 pharmaceutics-14-00280-f001:**
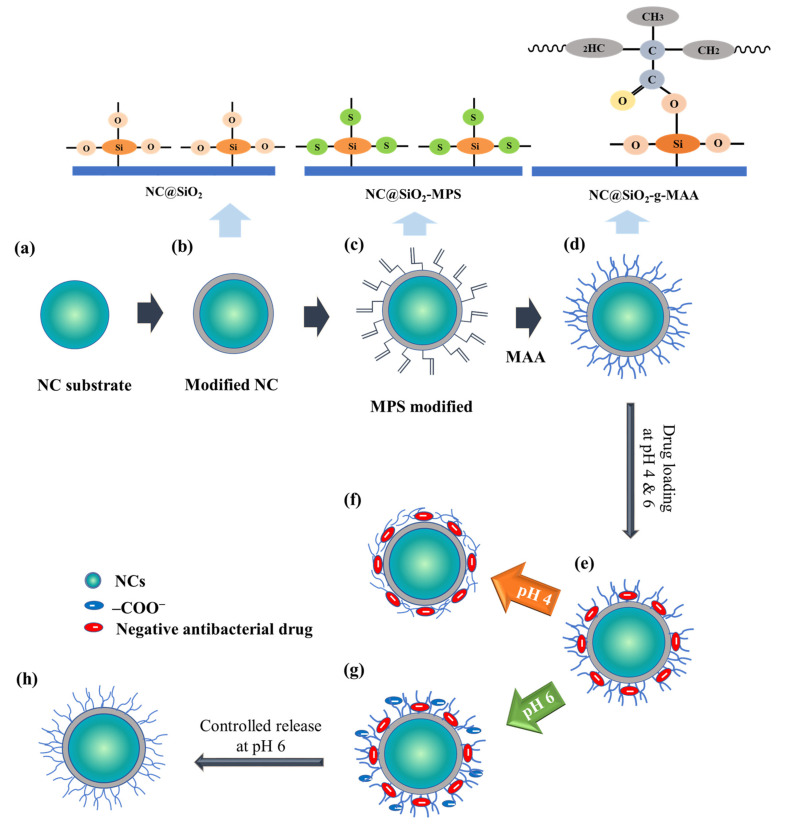
Chemical structure as it was modified to form pH- and humidity-sensitive self-antimicrobial NCs; (**a**) formation of NC substrate, (**b**) TEOS modification, (**c**) surface modification with MPS, (**d**) MAA ligand brushes grafted on the NC surface, (**e**) drug-loaded NC, (**f**) controlled drug release at pH 4, (**g**) controlled drug release at pH 6, and (**h**) pH triggered drug release at pH 6.

**Figure 2 pharmaceutics-14-00280-f002:**
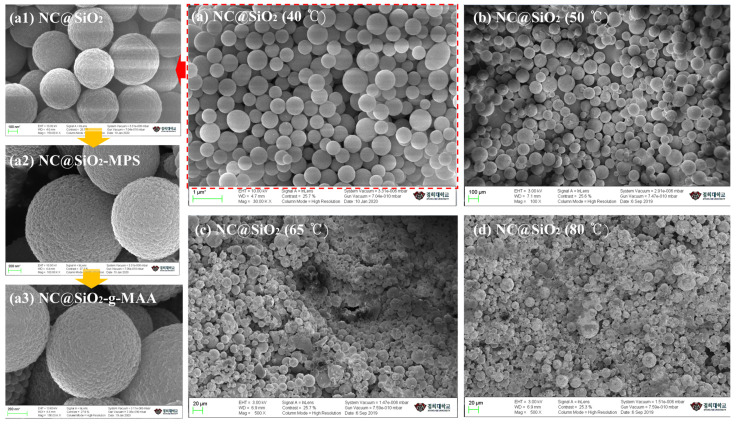
Aspects and shapes of NCs prepared at temperatures from 40 °C to 80 °C; (**a**) NC encapsulation at 40 °C, (**a1**) NC substrate, (**a2**) surface composition with MPS, (**a3**) ligand brush NC (NC@SiO_2_-g-MAA), (**b**) NC encapsulation at 50 °C, (**c**) NC encapsulation at 65 °C, and (**d**) NC encapsulation at 80 °C.

**Figure 3 pharmaceutics-14-00280-f003:**
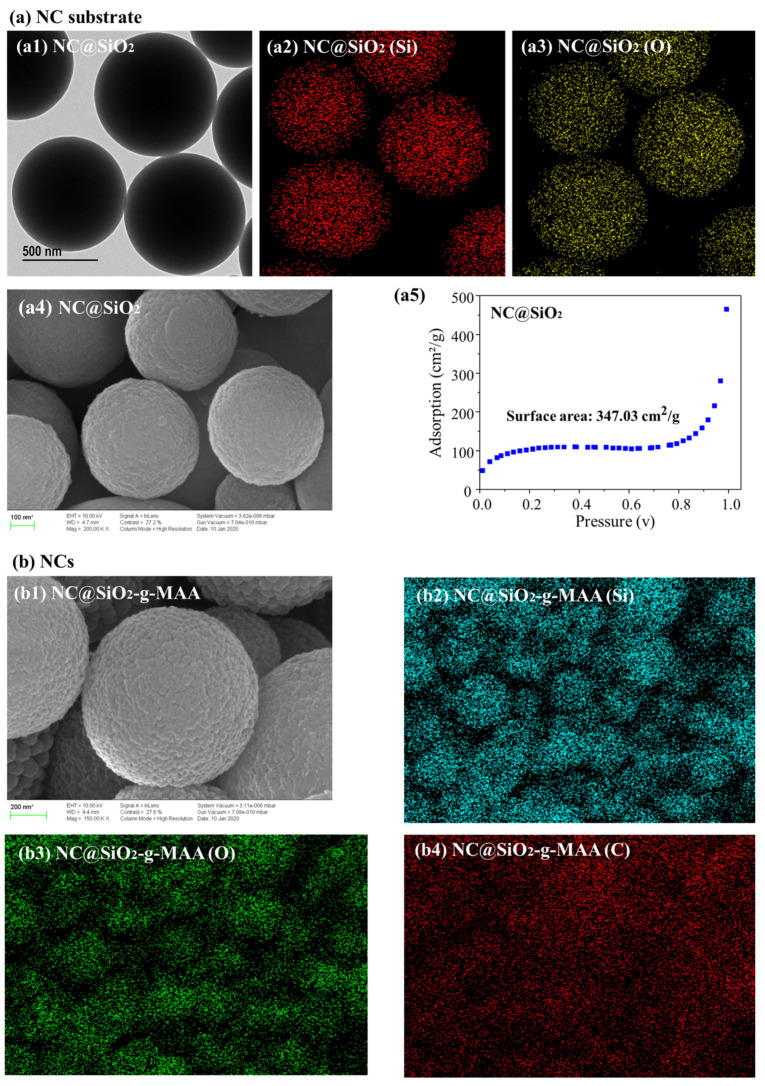
Analysis of morphology, surface shape, composition, and surface area of the (**a**) NC substrate and (**b**) self-antimicrobial NCs; (**a1**) shape analysis of the NC substrate (NC@SiO_2_) by TEM, (**a2**) Si group determination, (**a3**) O group determination, (**a4**) surface morphology by SEM, (**a5**) surface area by BET analysis, (**b1**) surface morphology by SEM on the NCs, (**b2**) Si group determination, (**b3**) O group determination, and (**b4**) C group determination.

**Figure 4 pharmaceutics-14-00280-f004:**
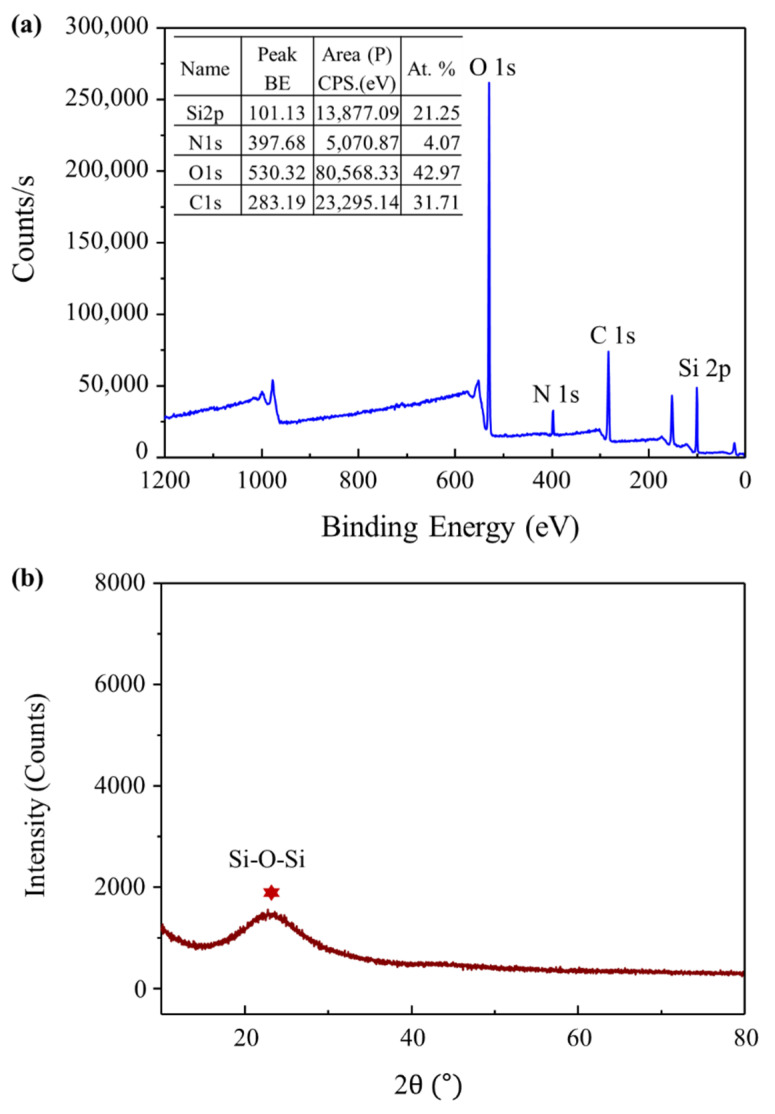
(**a**) XPS and (**b**) XRD analyses confirm the chemical component formation rate in the NCs.

**Figure 5 pharmaceutics-14-00280-f005:**
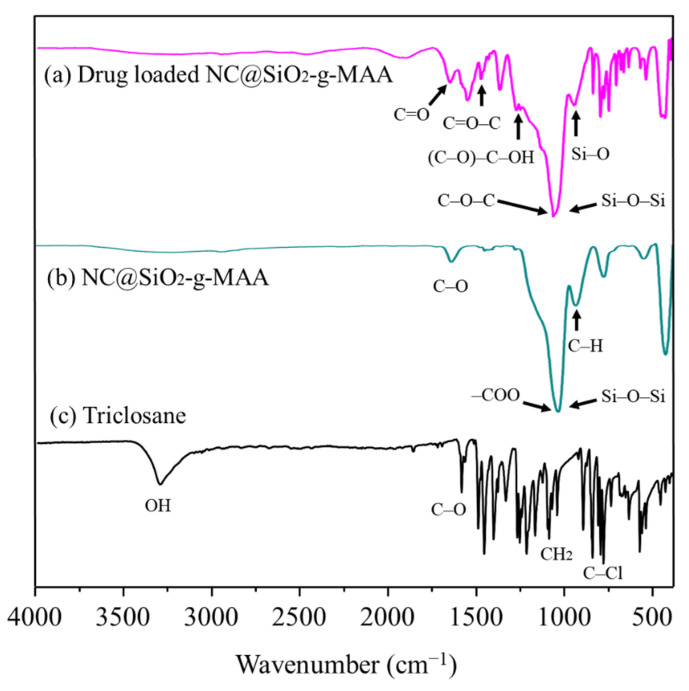
The FT-IR spectra for the (**a**) drug-loaded NCs, (**b**) pristine NCs and (**c**) triclosan (drug).

**Figure 6 pharmaceutics-14-00280-f006:**
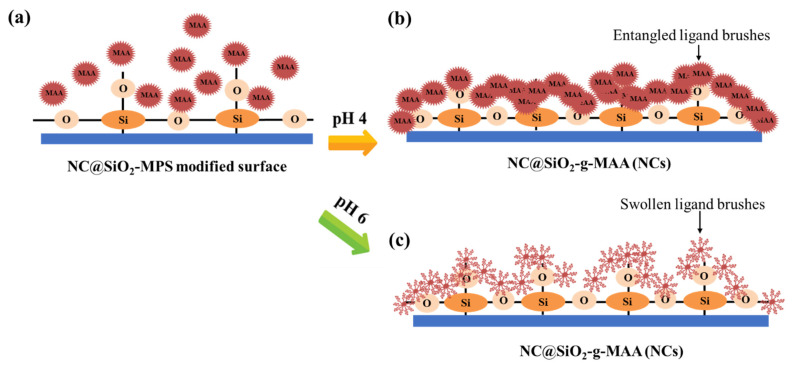
Self-controlled antibacterial drug release mechanism of the on the pristine NCs; (**a**) preparation of ligand brushes on the NC surface, (**b**) pH trigger mechanism of the ligand brushes at pH 4, and (**c**) pH trigger mechanism at pH 6.

**Figure 7 pharmaceutics-14-00280-f007:**
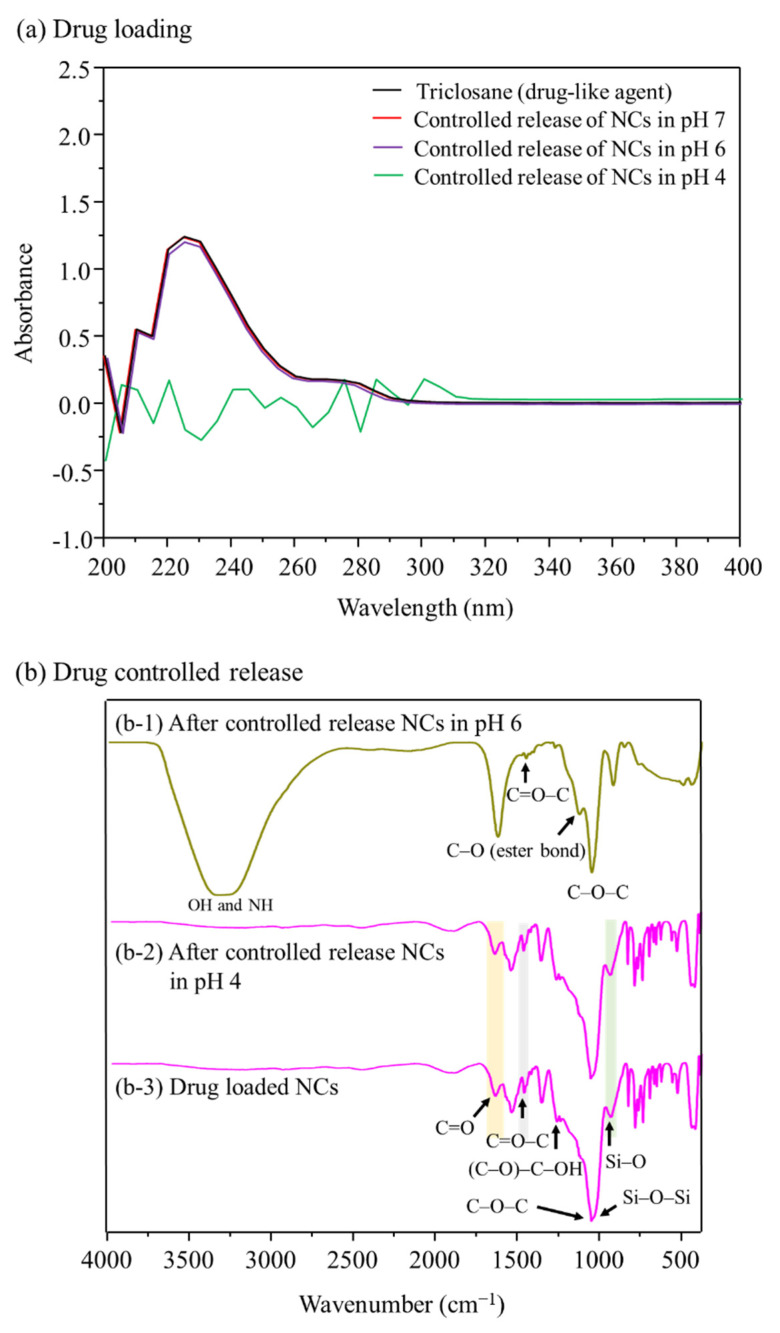
(**a**) Controlled release of triclosan from NCs at different pH values; (**b**) FT-IR spectra of the controlled-release NCs at different pH conditions: (**b-1**) drug-loaded NCs, (**b-2**) after controlled release at pH 4, and (**b-3**) after controlled release at pH 6.

**Figure 8 pharmaceutics-14-00280-f008:**
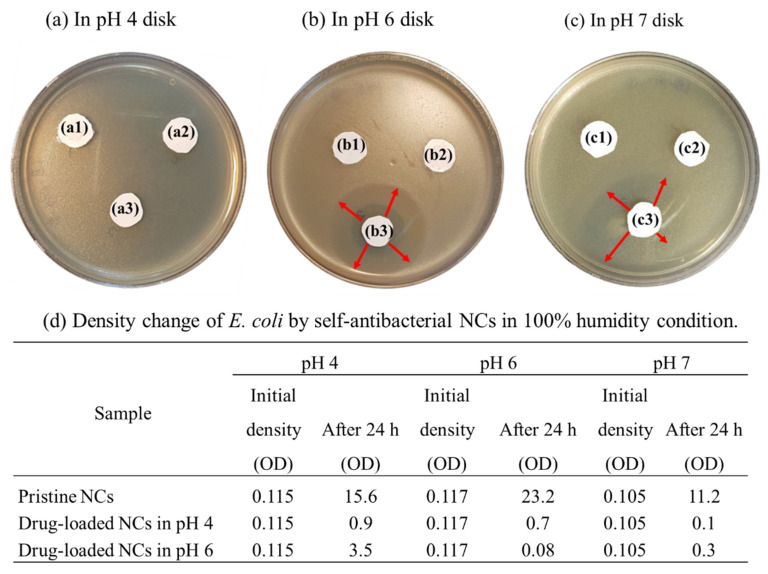
Self-antibacterial activity of ligand brush NCs at different pH conditions with 45% humidity of (**a**–**c**) pH 4, 6, 7 disks: (**a1**,**b1**,**c1**) control sample, (**a2**,**b2**,**c2**) pristine NCs, (**a3**,**b3**,**c3**) drug-loaded NCs in pH 4, 6, 7; (**d**) density change of *Escherichia coli*.

**Figure 9 pharmaceutics-14-00280-f009:**
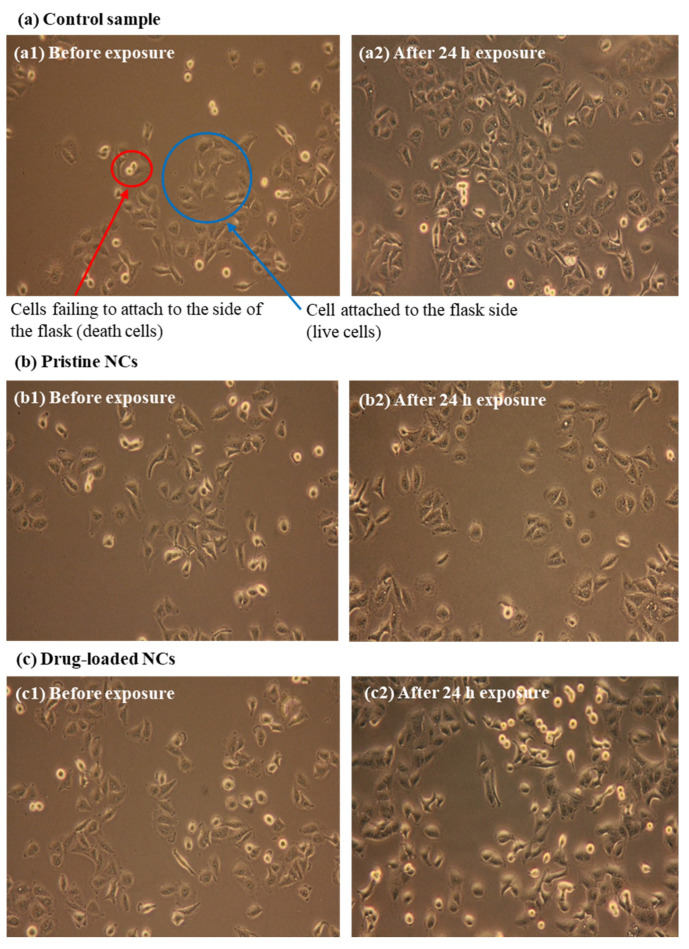
Antibacterial activity of ligand-brush NCs: (**a**) C.S., (**b**) pristine NCs, and (**c**) drug-loaded NCs.

**Table 1 pharmaceutics-14-00280-t001:** Zeta potential values and mean diameter of NCs at pH 7.

Sample	Zeta Potential [mV]	Mobility [cm^2^/Vs]	Diameter [nm]	Std. Dev. [nm]
NC@SiO_2_	−37.43	−2.92 × 10^−4^	676.3	119.6
NC@SiO_2_-MPS	−23.45	−1.86 × 10^−4^	761.2	100.1
NC@SiO_2_-g-MAA (NCs)	−36.41	−2.84 × 10^−4^	887.8	141.2

**Table 2 pharmaceutics-14-00280-t002:** The weight swelling ratio (q) of NCs in pH 4 and pH 6 conditions.

Buffer Solution	q Value of NCs							
	1 h	2 h	3 h	4 h	5 h	6 h	7 h	8 h
pH 4	1.000	1.002	1.007	1.009	1.011	1.014	1.015	1.015
pH 6	2.011	2.018	2.035	2.052	2.088	2.105	2.106	2.107

**Table 3 pharmaceutics-14-00280-t003:** Cytotoxicity of antibacterial NCs to cultured A549 cells.

Sample	Number of Cells Adhered		Cell Adhesion Activity (%)
	0 h	24 h	
Control	79	188	237.39
Pristine NCs	70	98	140.8
Drug-loaded NCs	115	162	140

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
