# Peer review of "Preparation of Ligand Brush Nanocapsules for Robust Self-Controlled Antimicrobial Activity with Low Cytotoxicity at Target pH and Humidity"

_pharmaceutics, 2022, doi:10.3390/pharmaceutics14020280_

Round 1

Reviewer 1 Report

A good manuscript. Just a couple of minor points:

-Title of section 3.3: change chemical potentials to zeta potentials. 

-Line 325: the pKa of a carboxlic group on a moinomoric acid liek acetic acid is in therange 4-5. With polymers, things are different. Each ionization wekaens the subseqeuent one resulting in abroadining of the titration curve and a higher average pKa. Please check the lietarture more thoroughly for an estimate of this average pKa of MAA.

Reviewer 2 Report

The work by E. Koh and Y. T. Lee, “Preparation of ligand brush nanocapsules for robust self-controlled antimicrobial activity with lower cytotoxicity at target pH and humidity” describe prepararion and characterization of some nanocapsules with self-controlled antimicrobial activity of triclosane under conditions of pH 6-7 and humidity 45-100%.

It is an interesting work but some certain issues need to be clarified.

Major concerns:

  1. Figure 8 is unclear. It seems to be incomplete since (c1) antibacterial drug-loaded NCs in pH 427 4, and (d1) drug-loaded NCs in pH 6 are missing from the figure.
  2. A clarification needs to be added because a very similar work has been published by Eunjoo Koh, Yong Taek Lee, Development of humidity and pH responsive ligand brush porous nanocapsules for self-controlled antibacterial properties without cytotoxicity, https://doi.org/10.1016/j.apsusc.2021.150133.
  3. English needs to be revised and improved.

Minor points:

In 2.1 Materials,

polyvinylpyrrolidone in mentioned twice

line 76: poly(methalic acid, sodium salt) solution, instead of methacrylic acid

Reviewer 3 Report

In this study, a nanocapsule with controlled release of drugs under the target pH and humidity was synthesized, and the capsule has a good bactericidal effect on Escherichia coli. And the nanocapsule has good biocompatibility to A 549 cells. The nanocapsule has a good application prospect in the medical and health field, and has certain reference significance for the controlled release of loaded drugs. However, there are still some errors in this article. The suggestions are as follows:

  1. There are many formatting errors in the text, for example, the name of the bacteria in the header and legend in Fig. 8 is not italicized; the format in the table in Fig. 4 is inconsistent with the format in the figure, etc.;
  2. The subtitle of Chapter 2.3 is incorrect, and there is a format error in the chapter;
  3. Only for the evaluation of the antibacterial effect of the large intestine, is there no golden grapes? Adding some clinically isolated strains can explain the problem and make more sense;
  4. What is the snowflake-like substance in Fig. 6c? Please give the logo;
  5. Why use triclosane, an anionic drug, so that the zeta potential of the nanocapsules is negatively charged after modification, while the bacterial surface is mostly negatively charged, why not consider using a cationic drug? This may have a better effect on bacteria;
  6. About why the OD value in Fig. 8 is as high as 15.6. The OD value should not be zero. Even if all the bacteria die, there will be an OD value. The OD value measures the turbidity of the bacteria, not the number of viable bacteria. Are there any parallels?
  7. Some statements in the text are too absolute, for example, non-toxic to cells should be written as having good biocompatibility;
  8. It is recommended to give some prospects for the application of the nanocapsules.

Reviewer 4 Report

The manuscript needs a huge reformulation.

  • The objective of the research is not well explained and data presented is very confused.
  • The developed NCs presented in the present manuscript are to be applied to what ? Industrial application?
  • In the introduction section the authors refer “The size of the NCs was controlled to no more than 0.1 μm to ensure proper surface area” – Are you sure this sentence is in accordance to data presented ?
  • The Section 2.4. "Cytotoxicity test of lung epithelial cells" should be clearly rewritten. Here are some examples:

What do the authors mean with “The sample was diluted to a certain concentration”?

“Dispensed cells were cultured for 24 h, the sample was exposed to the exposing material/solution for 5 min under sonication” - - Which sample ?

“Three exposing liquids” - What do the authors mean?

How was the cellular viability evaluated?

  • In methods description when a centrifugation step is used “rpm” should not be used. Instead “rpm” , g force should be used.
  • Data presented in Figure 8 was repeated how many times? How many replicates were used in each experimental assay? What does it mean an OD of 15.6 or 23.2 in Figure 8 ?
  • The English should be revised in all document.

Round 2

Reviewer 2 Report

The revisions have improved the manuscript. The authors answered to the questions raised by the reviewer.

However, there are still some minor points which need to be addressed before publication.

There are two figures for fig 2.

The 1st version of the fig 8 should be removed.

Reviewer 3 Report

My concerns have been addressed in the revised version, the manuscript is now acceptable.

Reviewer 4 Report

  • A huge reformulation in the manuscript was performed. Nevertheless, some sentences still need to be reformulated.

Here are some examples:

Lines 315-317 - ”The mean diameter of the capsule thus increased with the number of processes by which it was reformed, which was affected by hydrogen bonding and produced the grafted brushes on the NC surface”

Lines 356-359 “We analyzed the controlled release performance of drug-loaded NCs at pH 4, pH 6, and pH 7. and found that controlled release of the antibacterial drug occurred only at pH 6. “

Lines – 531-532 “This work has demonstrated humidity- and pH -responsive, hybrid NCs prepared with ion changeable polymers that react at the target pH.”

  • In Figure 9c) replace “drag loaded NCs” by “drug loaded NCs”
  • In methods section several centrifugation conditions are still in rpm
  • In section 2.5 replace “Smoluchows” by “Smoluchowski”
